# Disinfection Efficacy of Slightly Acidic Electrolyzed Water Combined with Chemical Treatments on Fresh Fruits at the Industrial Scale

**DOI:** 10.3390/foods8100497

**Published:** 2019-10-14

**Authors:** Xiuqin Chen, Charles Nkufi Tango, Eric Banan-Mwine Daliri, Seong-Yoon Oh, Deog-Hwan Oh

**Affiliations:** 1Department of Food Science and Biotechnology, College of Agriculture and Life Sciences, Kangwon National University, Chuncheon 200-701, Korea; cxq20135331@gmail.com (X.C.); Charlynkufi2@yahoo.fr (C.N.T.); ericdaliri@yahoo.com (E.B.-M.D.);; 2Division of Cancer Epidemiology and Management, Center for Uterine Cancer, National Cancer Center, Ilsandong-gu, Goyang 410-769, Korea; 3Three and Four Bio, 882-1 Jusanri Hojeomeyon, Wonju, Kangwon-do 109-111, Korea

**Keywords:** slightly acidic electrolyzed water, combined treatments, inactivation, fruits storage

## Abstract

The objective of this study was to investigate the efficacy of slightly acidic electrolyzed water (SAEW) combined with fumaric acid (FA) and calcium oxide (CaO) treatment on the microbial disinfection of fresh fruits including apple, mandarin, and tomato at the industrial scale. The combined treatments can significantly (*p* < 0.05) reduce the population of natural microbiota from the fruit surfaces and the treated samples showed good sensory qualities during refrigeration storage. In addition, decontamination of inoculated foodborne pathogens (*Escherichia coli* O157:H7 and *Listeria monocytogenes*) was carried out in the laboratory, and the combined treatments resulted in a reduction ranging from 2.85 to 5.35 log CFU/fruit, CaO followed by SAEW+FA treatment that resulted in significantly higher reduction than for SAEW+FA treatment. The technology developed by this study has been used in a fresh fruit industry and has greatly improved the quality of the products. These findings suggest that the synergistic properties of the combination of SAEW, FA, and CaO could be used in the fresh fruit industry as an effective sanitizer.

## 1. Introduction

The consumption of fresh fruits has significantly increased over the last decade [1]. This increase could be due to heightened public awareness of the benefits of fresh food consumption such as providing the body with vitamin A, vitamin C, folate, minerals, dietary fiber, and phytochemicals [2,3,4]. Korea imported 834,000 tons of fruit, worth $1.24 billion in 2017. And this marked the highest number ever, according to the data compiled by the Korea Customs Service. Fresh fruits such as apple, mandarin, and tomato are minimally processed and are typical commercial fruits in Korea, Japan, and China [4,5]. However, there are some risks associated with consuming fresh fruit. Pesticide residues on fresh fruit pose health hazards to human beings [6]. Also, fruits serve as key vectors for foodborne pathogens. The rate of foodborne illness caused by the consumption of fresh fruits remains high [7], and the USA Center for Disease Control has reported that 2420 foodborne outbreaks between 2009 to 2015 were associated with consumption of contaminated fruits. Providing consumers with safe fruits can greatly increase their recommended intake of vitamin A, vitamin C, folate, minerals, dietary fiber, and phytochemicals per day, so as to improve their health [8]. Therefore, postharvest technologies need to be developed to control the microbiological quality of fresh fruits.

Developing good postharvest handling techniques and technologies to extend shelf-life of produce and minimizing the survival of foodborne pathogens is the most important step in the food industry [7,9]. Over the years, a variety of decontamination technologies have been used to increase the safety of fresh-cut fruits [10,11]. For instance, treatment of contaminated spinach, sesame leaf, and cabbage with chlorine-based-sanitizers exhibited ≥2.0 log CFU/g reductions in the microbial load [12,13]. However, the application of traditional chlorine-containing disinfectants in the food industry has many limitations since it causes highly adverse effect to human health and environment. For this reason, acidic electrolyzed water (AEW) has been developed as an environmentally friendly antimicrobial agent as a substitute for chlorine-based-sanitizers [14]. David Santo et al. tested the antibacterial activity of AEW in the inhibition of *Escherichia coli* (*E. coli*) and *Cronobacter sakazakii* (*C. sakazakii*) on fresh-cut mangoes; the results showed that AEW resulted in declining of *E. coli* (1.96 log CFU/g) and *C. sakazakii* (1.76 log CFU/g) populations [15]. Since fresh-cut fruits were a suitable substrate for the survival and growth of foodborne pathogens, the effect of AEW on foodborne bacteria population of fresh-cut fruits was demonstrated though several studies, such as *E. coli*, *Salmonella enterica,* and *Listeria spp* inoculated on ‘Rocha’ fresh-cut pears decreased values of 0.53–1.1 log CFU/g were achieved by AEW (100 mg/L of free chlorine) washings [16]. Besides, AEW at 50 and 100 mg/L of free chlorine were used to treatment apple slices inoculated with *E. coli*, *Listeria innocua,* or *Salmonella choleraesuis* and significantly decreased the populations of pathogen, when compared to that of sodium hypochlorite solution and distilled water [17].

AEW is considered as one potential substitute for traditional chlorine treatment. It can be generated by electrolysis of saline solution, generally sodium chloride (NaCl) and/or hydrochloric acid (HCl) mixed with tap water in an electrolytic cell [18,19]. When the electrolytic cell does not contain a membrane, it results in slightly acidic electrolyzed water (SAEW; pH 5.0–6.5) [20]. SAEW usually shows higher antimicrobial activity and have been recognized as a novel, environmental-friendly, highly effective antimicrobial agent [21]. Studies have shown that the systematic application of SAEW could reduce the natural microbiota of buckwheat sends and sprouts [22]. Another study found that SAEW also has sufficient efficacy in removing pesticide residues and killing some insect pests existing on fruits without compromising sensory and nutritional quality [23].

However, SAEW will lose its antiseptic effect quickly when exposed to air and light, and the reduction of Gram-negative foodborne pathogens on food from SAEW individual treatment is limited. In recent years, more and more researchers study enhancing SAEW antimicrobial efficacy using other sanitizer decontaminants to ensure produce safety [24]. Ding et al. [14] tested the combination effect of ultrasonic and SAEW on microbial loads and quality of cherry tomatoes and strawberries; the results showed that ultrasonic treatment can improve the disinfection effect of SAEW with 1.77 and 1.29 log reduction of total aerobic bacteria (TAB) on the surface of cherry tomatoes and strawberries. Besides, there are studies on the inactivation efficacy of SAEW in combination with both physical and chemical treatment. Hussain et al. observed that benzalkonium chloride and mild heat treatments combined with SAEW reduced *Bacillus cereus* biofilm by 2.62 log CFU/cm^2^, much higher than when only SAEW treatment was used [25].

Many studies have been performed to identify fumaric acid (FA) and calcium oxide (CaO) that can be used to improve the microbial safety of fresh produce as antimicrobial agents. FA is a food grade acidifier with strong bactericidal activity. A combined treatment of FA and safflower seed meal extract can be an effective sanitizing method to improve the microbiological safety of fresh lettuce without affecting its quality [26]. Chitosan was modified with FA to enhance its solubility and antibacterial activity and fewer harmful compounds are formed by reactions [27]. CaO has been used as a natural antimicrobial agent for inactivating the pathogens in fresh produce for several years [28,29]. The antimicrobial activity of combined treatment with aqueous ClO_2_ and CaO was investigated. CaO showed high activity on bacterial inactivation in fresh-cut kale [30].

Some pathogen-inactivating technologies, which have different bactericidal targets, were used to treat food; the synergistic effect of those technologies is defined as minimizing deterioration of food quality by reducing treatment intensity and time. In this study, a hurdle approach that combines FA and CaO treatment to improve the antimicrobial effect of SAEW against pathogens as well as prolong shelf-life and the quality of fresh fruits were developed and performed in an industry that processes fresh fruits.

## 2. Material and Methods

### 2.1. Inactivation Efficiency of Natural Microbiota

#### 2.1.1. Samples Preparation for Treatment

Fresh apples, mandarins, and cherry tomato fruits were provided by Nette company suppliers at Chuncheon city, South Korea at commercial maturity. These samples were directly transported to the Nette Corporation using a refrigerated truck. Each kind of fruit was separately weighed (2 kg) and packaged in polyethylene bags and stored at 4 °C to use for not more than 24 h before the experiment. Each polyethylene bag was considered as a batch, and three batches per fruit were used in this study.

#### 2.1.2. Sanitizer Preparation

In this study, the electrolyzed water generator was developed by assembling a water tank, electrolytic cell, power supply, and master flex according to the principal of SAEW generation. SAEW were generated by electrolysis of a combined solution of dilute hydrochloric acid (5%, Sigma-Aldrich, St- Louis, MO, USA) and sodium chloride (2M, Sigma, USA) called electrolyte. The electrolyte flow passed through a electrolytic cell at a rate of 2 mL/min, at setting of current 3.8–3.9 V, amperage 10 A, the rate of electrolyte flow is controlled by master flex, and the electrolyte diluted with tap water storage in the water tank at a flow rate of 4 L/min using a pump (Barnant masterflex tube, Barnant Co., Massachusetts, IL, USA) to generate SAEW. SAEW was collected after the generator worked more than 30 min, in order to make sure that a stable current and amperage were reached. The initial concentration of the available chlorine (ACC) in the SAEW reached 30 ppm, the oxidation reduction potential (ORP; 818–854 mV) was measured with an ORP meter (HM-60V, TOA Electronics Ltd., Tokyo, Japan), and pH of SAEW (5.42 ± 0.15) was measured using a pH meter (D-22, Horiba, Kyoto, Japan). The SAEW were collected at low temperature (4 °C) and used rapidly in the experiments before the properties changed. Crystalline fumaric acid (FA) and calcium oxide (CaO) used in this study were provided by Eco-Biotech Company (Hwaseong-Si, Gyeonggi-Do, Korea). FA was dissolved in sterile distilled water to give a diluted solution with concentration of 0.5% (^w^/^v^) with a pH of 2.38 ± 0.19 while the manufactured CaO solution was prepared at a concentration of 0.2% (^w^/^v^) and pH 12.09 ± 0.36.

#### 2.1.3. Disinfection Treatment Procedures

Industrial-scale experiments were performed to assess the decontamination impact of three different fresh fruits in this study. A sequential treatment line (Figure 1) including CaO treatment, followed by FA and SAEW, was developed at Nette company (Chuncheon, South Korea).

#### 2.1.4. Natural Microbiota Microbiological Analysis

One sample of each fruit was aseptically mixed with 10 mL D/E neutralizing broth (D/ENB, Difco) in individual Whirl Pack bags (Nasco Whirl-Pak, Janesville, WI, USA). Subsequently, the samples were vigorously shaken followed by rubbing and massaging for 5 min to suspend surface microorganisms. After homogenization, 1 mL of sample suspension containing natural microbiota were mixed with Tryptone Soy agar (TSA Difco) and poured into plate to enumerate the total aerobic bacteria (TAB). Dichloran Rose Bengal Chloramphenicol (DRBC Difco) agar were used to enumerate the yeast and mold. Then, 1 mL of sample suspension were 10-fold diluted with 0.1% buffered peptone water (BPW Difco). Population of cell in each culture was confirmed by plating 0.1 mL serial dilution on the agar plates and incubating at 37 °C for 24 h. Population of total coliforms was confirmed by plating 1 mL serial dilution on the Petrifilm™ Coliform Count Plate (3M Center, St. Paul, MN, USA). Colonies were enumerated and expressed as log CFU/fruit, and the detection limit for microbiological counts is 1 log CFU/fruit.

#### 2.1.5. Survival of Natural Microbiota during Storage

Two kilograms of fresh fruits were packaged using a stomacher bag and kept at 4 °C after the decontamination procedure as described in Section 2.1.3 to determine the effect of hurdle technology on the survival of natural microbiota during 14 days of storage. One sample of each fruit was randomly chosen every 48 h and packed in the individual Whirl Pack bags. Each fruit were mixed with 10 mL D/ENB in the bags and vigorously shaken followed by rubbing and massaging for 5 min to suspend surface microorganisms to the broth, and population of cell in each culture were confirmed as described in Section 2.1.4. The growth date of TAB, yeast and mold, and total coliforms during storage were monitored. Unwashed fruits were used to evaluation the capacity of the treatments the influence the shelf-life.

### 2.2. Inactivation Efficiency of Pathogenic Bacteria on The Surface of Fruits

#### 2.2.1. Bacterial Culture

The experiment was carried out in the laboratory of Food Microbiology and Safety, Kangwon National University, Chuncheon, Korea. *Escherichia coli* O157:H7 (*E. coil* O157:H7) and *Listeria monocytogenes* (*L. monocytogenes* ATCC 19118, Scott A) were the pathogenic bacteria used to contaminate fruits. Each suspension (0.1 mL) of the stock cultures was plated on the selective agar media individually. MacConkey Sorbitol Agar, which supplemented crystal violet and bile salt mixture (MSA Kisan Bio Co. Ltd., Seoul, South Korea ) and Oxford base medium agar supplemented with modified Oxford antimicrobic supplement (Difco) (OBMA, Difco) were used as selective media for the growth of *E. coil* O157:H7 and *L. monocytogenes*, respectively. After incubation for 24 h at 37 °C, a single colony was transferred from the selective agar plate into 10 mL of TSB and incubated at 37 °C for 18–24 h. Following the incubation, the bacteria suspension (0.1 mL) was transferred into 10 mL of TSB and incubated at 37 °C for 18–20 h. The overnight culture was centrifuged at 4000× *g* for 10 min and the cells were washed three times with 0.1% BPW, and the final cell concentration was adjusted to 10^8–9^ CFU/mL with 0.1% BPW.

#### 2.2.2. Procedure of Samples Preparation and Inoculation

The fresh apple, mandarin, and tomato fruits were purchased from supermarkets at Chuncheon city, South Korea. The samples were directly transported to the laboratory using a refrigerated truck and they were stored at 4 °C until pathogenic inactivation experiment. Cell suspensions (150 mL) of each strain prepared at Section 2.2.1 were mixed in a sterile conical flask to obtain 300 mL cocktail; following this, 300 mL cocktail containing two pathogenic bacteria was added into 700 mL 0.1% BPW to obtain 1 L cell suspension. Fresh fruits were dipped in the 70% ethanol for 1 min before inoculation, in order to eliminate the interference of natural microbiota on the surface of fruits. Sterile distilled water (DW) was used to rinse the disinfected fruits and remove the remaining ethanol residue and then dried in a laminar flow safety cabinet. The ultra-violet lamp of laminar flow safety cabinet was kept on at all times to make sure the fruits were not contaminated with other bacteria during the period of desiccation. Each kind of fruit was separately weighed (2 kg) and packaged in polyethylene bags as batches. Each batch was inoculated with pathogenic bacteria evenly by dipping in microbial cocktail suspensions (1 L) for 10 min. Treated fruits were left in a laminar flow safety cabinet for approximately 1 h until the liquid on the surface of fruits were evaporated.

#### 2.2.3. Decontamination Procedure

Three treatment lines were established for the inactivation experiment. Treatment tanks filled with 6 L of sanitizer solution (CaO, FA+SAEW, and tap water, respectively) were placed in each line. In the first line, inoculated samples were washed sequentially in 6 L of CaO, FA+SAEW, at room temperature (RT, 23 ± 0.15) for 3 min followed by washing with Tap water (TW) for 2 min. For the second line, the inoculated samples were washed with TW, FA+SAEW sanitizer for 3 min followed by washing with TW for 2 min. In the third line, the samples were washed twice with TW (for 3 min in the first tank followed by for 2 min in second tank) as control.

#### 2.2.4. Microbiological Analysis

Two fruits of each batch were randomly chosen after the decontamination experiment and packed in the individual Whirl Pack bags. The samples were mixed with D/ENB in the bags and vigorously shaken followed by rubbing and massaging for 5 min to suspend surface microorganisms to the broth. The populations of surviving *E. coil* O157:H7 and *L. monocytogenes* enumerated using selective media MSA and OBMA, respectively. All the plates were then incubated for 24–48 h at 37 °C, and the number of colonies were expressed as log CFU/fruit.

### 2.3. Storage Test

Fruit samples were packaged using a stomacher bag and stored at 4 °C after chemical treatments. In order to determine the effect of hurdle technology on the quality of fruits, the fruits quality (overall visual appearance, off-odor, and texture) and survival of natural microbiota on the surface of fruits were monitored during storage for 14 days. The end of shelf-life was considered as the time when the bacterial and fungi population reached 7 and 5 log, respectively [31].

Generally, the evaluation of fresh fruit sensory quality was performed on item qualities such as appearance, color, flavor and texture, and overall visual defects [32]. The end of sensory shelf-life of fruits was due when the evaluation parameters dropped below the acceptable levels [33]. A trained panel of six students (age range 20–30) in a room under daylight with separate booths evaluated the sensory quality of the treated fruits. The qualities were scored by the numeric scale 9 = excellent, 5 = fair, 1 = extremely poor for OVQ (overall visual quality) and 5 = no off-odor, 3 = acceptable, 1 = severe for off-odor [25].

### 2.4. Statistical Analyses

Each experiment was performed in duplicate and mean values for all indicators were calculated from the independent triplicate trials. The log reduction on bacterial population was calculated and results were subjected to one-way analysis of variance (ANOVA) using SPSS statistics software version 21 (SPSS Inc., IBM Company, Chicago, IL, USA). Differences between means were determined using Tukey’s multiple comparison test and a significant difference was considered at *p* < 0.05.

## 3. Results and Discussion

### 3.1. Inactivation Efficiency of Natural Microbiota of Different Treatments on Fresh Fruits

This study was designed for the fruit process industry with the purpose of improving the quality of fruit products; the experiment was carried out under industrial scale.

#### 3.1.1. Inactivation of Total Aerobic Bacteria (TAB) on Fruits with Different Treatments

Both FA+SAEW and CaO-FA+SAEW treatments significantly reduced the population of TAB on the surface of apple, mandarin, and tomato, ranging from 2.31 to 4.08 log CFU/fruit (Figure 2A) compared to the control treatment (TW). This is in agreement with the results presented by other researchers that SAEW, FA, and CaO were highly effective on the bacterial reduction as chemical sanitizers [18,24,29]. The mean initial populations of TAB on the fresh apple, mandarin, and tomato fruits were 5.02, 4.72, and 5.98 log CFU/fruit, respectively. Treatment of CaO-FA+SAEW resulted in a higher reduction for microbial population of TAB on the surface of fruits when compared to the FA+SAEW; the results may be explained by the fact that CaO has a pH of about 12.5, which can be associated with antimicrobial activity by destroying the outer membrane and interfering in the microbial enzymatic activity [34]. In addition, it can control physiological disorders and remove residual pesticide and other debris during postharvest processing. It has been reported that calcium ion nanomaterials show high performance in improving the inactivation of microbes [35]. The highest reduction was observed for apples treated with FA+SAEW and CaO-FA+SAEW with the counts of TAB reduced by 3.94 and 4.08 log CFU/fruit, respectively. Results of the combination of three sanitizer treatment (CaO-FA+SAEW) in our study were a little higher than that reported by Tango et al. who observed a reduction of 3.24 ± 0.32 log CFU/fruit for TAB on apple after treatment with CaO-FA+SAEW [3]. The difference between two results may be explained by the instability of SAEW when exposed to air and light, which may influence its effectiveness in the bacterial reduction [14]. The TAB reduction of different fruits after combined treatments could be ranked as following: Apple > mandarin > tomato.

#### 3.1.2. Inactivation of Coliforms on Fruits with Different Combined Treatments

Results of disinfection effect of FA+SAEW and CaO-FA+SAEW treatments for coliform on the surface of fruits are shown in Figure 2B. The initial populations of coliforms on the fresh apple, mandarin, and tomato fruits were 3.93, 4.24, and 5.96 log CFU/fruit, respectively. FA and SAEW synergistically increased the disinfection efficiency against coliforms compared to TW, especially on the surface of mandarin where the reduction count was 4.24 log CFU/fruit. The higher reduction of coliform populations on the surface of mandarin may be explained by the fact that food surface properties such as hydrophobicity, electric charge, and roughness may influence the adhesion of microbial [16]. In addition, there was no significant difference (*p* > 0.05) between the reduction of coliforms on mandarin fruits washed with CaO and those washed without CaO. This may be due to peel pitting disorder in mandarin fruit, which can affect the sterilization efficiency of CaO [36]. When apple and tomato fruits were treated with FA+SAEW, the coliform populations were reduced by 2.83 ± 0.31 and 2.42 ± 0.04 log CFU/fruit compared to 3.49 ± 0.05 and 3.02 ± 0.05 log CFU/fruit observed on apple and tomato fruits treated with CaO-FA+SAEW.

#### 3.1.3. Inactivation of Yeast and Mold on Fruits with Different Combined Treatments

Treatment with FA+SAEW for 3 min caused 3.24, 2.47, and 2.10 log reductions of yeast and mold inoculated on the surface of apple, mandarin, and tomato fruits, respectively, while treatment of CaO-FA+SAEW caused 4.27, 2.87, 2.35 log reductions (Figure 2C). Similarly, total combined treatment showed additive effects in the inactivation of yeast and mold, especially on the surface of apple fruits.

### 3.2. Effect of Combined Treatment on Natural Microbiota Growth During Storage of Fruits

The behavior of TAB on the surface of apple, mandarin, and tomato fruits was monitored during storage at 4 °C. The changes in the population of TAB on the treated apple during 14 days of storage are shown in Figure 3A. The population of TAB on the surface of apple treated with FA+SAEW and CaO-FA+SAEW was reduced to 1.08 ± 0.00 and 0.94 ± 0.71 log CFU/fruit, respectively. After 14 days of growth, they reached 3.78 ± 0.42 and 2.96 ± 0.63 log CFU/fruit, respectively. TAB population of fruits washed with TW were higher compared to those washed with combined sanitizer. The results presented indicate that FA and SAEW can inhibit the growth of TAB on the surface of apple, mandarin, and tomato during storage at 4 °C. The end of shelf-life of a fresh produce is considered when TAB reaches the maximum acceptable level of ≥7 log CFU/fruit [3], while the population of TAB on unwashed fruits reached 7 log CFU/fruit at the end of the storage.

Natural microbiota on the surface of untreated (control) and TW washed mandarin fruits increased rapidly compared to those that grew on FA+SAEW and CaO-FA+SAEW treated mandarin fruits (Figure 3B, Figure 4B and Figure 5B). The counts of TAB on mandarin after treatment with FA+SAEW and CaO-FA+SAEW were reduced to 1.61 ± 0.47 and 0.98 ± 0.03 log CFU/fruit, respectively. After 14 days of storage at 4 °C, there was a significant difference (*p* < 0.05) in the final population of TAB between treated mandarin and those untreated (Figure 3B). Results from the present study demonstrated that the TAB of the untreated (control, TW) and treated groups (FA+SAEW, CaO-FA+SAEW) during storage at 4 °C did not reach the unacceptable levels (≥7 log CFU/fruit) within 14 days. Similarly, the population of coliform on mandarin still remained within acceptable limits throughout the storage period (Figure 4B). When compared with the other two fruits, the mandarin has a longer shelf-life and the growth rate of natural microbiota on the mandarin surface were slower; this can be explained by the mandarin fruit skin citrus containing acid and essential oil, which may inhibit the growth of bacteria. These results can help explain why, still now, the mandarin has been less reported as associated with foodborne pathogens diseases when compared with the other two fruits.

For tomato fruits, FA+SAEW, CaO-FA+SAEW resulted in a high reduction of TAB, coliforms, and yeast and mold. Microorganisms on the surface of tomato fruits increased rapidly and reached the shelf-life limit after 14 days of storage at 4 °C. No significant (*p* > 0.05) differences were observed between FA+SAEW and CaO-FA+SAEW treated tomato fruits in relation to the levels of TAB. Also, bacteria populations on samples subjected to these treatments exceeded the unacceptable levels after 12 days (Figure 3C). Coliforms levels (Figure 4C) as well as yeast and molds levels (Figure 5C) for tomato fruits treated with FA+SAEW and CaO-FA+SAEW also exceeded the unacceptable level after 12 days. Yeast and mold level for tomato fruits treated with FA+SAEW and CaO-FA+SAEW exceeded the unacceptable level (≥5 log CFU/fruit) after six days and nine days, respectively (Figure 5C).

### 3.3. Impact of Hurdle Treatments on Sensory Quality Changes of Fresh Fruit during Storage at 4 °C

During storage, changes in quality of produce may occur due to physicochemical stress caused by hurdle treatment (SAEW + FA, CaO-FA+SAEW). Sensory attribute results of the treated fresh apple, mandarin, and tomato fruits are summarized in Table 1, Table 2 and Table 3, respectively. For all the treatments, there was no significant change (*p* > 0.05) on the scores of appearance, color, off-odor, and overall quality after 12 days of refrigeration storage. The results were in agreement with Daniel Rico et al., which the panelist considered acceptable all the lettuce samples treated with EW (12, 60, 120 mg/L of free chlorine) during seven days of storage [37]. Several studies reported that EW did not affect the quality parameters of fresh production. In a study conducted by Thi-Van Nguyen et al., EW (20, 60 mg/L of free chlorine) was effective as a disinfectant for fresh-cut baby spinach and remained above acceptable levels over 13 days of storage at 4 °C [38]. However, after 14 days of storage, the sensory quality of untreated and treated apple changed significantly except off-odor. The sensory quality results of treated tomato are summarized in Table 3. The scores of appearance, color, off-odor, and overall quality of tomato significantly reduced with increasing storage time relative to apple and mandarin.

The quality of unwashed and TW washed apples was decreased when compared to the treated apple (Table 1). CaO treatment can help to improve the sensory quality of fruit probably by improving the hardness of the fruit and peroxidase activity [39]. The results are in agreement with Section 3.2 and the behavior of microbes during refrigeration storage. Combined treatment showed additive effects in the inactivation of microbes and prolongs the shelf-life of fresh fruits.

### 3.4. Foodborne Pathogens Efficiency of Different Treatments on Fresh Fruits

Samples were inoculated with *E. coli* O157:H7 and *L. monocytogenes*. The initial populations of *E. coli* O157:H7 on the surface of apple, mandarin, and tomato fruits used in this study were 6.78 ± 0.14, 6.18 ± 0.35, and 6.23 ± 0.15 log CFU/fruit, respectively. The reduction in numbers of *E. coli* O157:H7 on fruits during SAEW + FA and CaO-FA+SAEW is presented in Figure 6A. SAEW+FA resulted in 3.16 ± 0.49, 3.14 ± 0.07, and 2.85 ± 0.06 log reductions of *E. coli* O157:H7 on apple, mandarin, and tomato fruits, respectively. The results described in this study are different to previous research regarding reductions of *E. coli* on fruit. Danyluk et al. [40] reported that 200 ppm free chlorine of SAEW wetting stem scar of grapefruit resulted in 4.93 log CFU/g reductions of *E. coli* on the surface of grapefruit. The higher reduction of *E. coli* may be explained by that higher available free chlorine concentration of SAEW were used in their study. Santo et al. [15] also studied the efficacy of electrolyzed water to kill *E. coli* on processed mangos, and the results showed that there was a significant reduction when washed with neutral electrolyzed water and sodium hypochlorite (2.2 log CFU/g). In our study, available free chlorine concentration of SAEW was 30 ppm. CaO-FA+SAEW treatment showed additive effects in the inactivation of *E. coli* O157:H7 on mandarin by 4.80 ± 0.07 log CFU/fruit. For *L. monocytogenes*, the initial populations on fruit surfaces (apple, mandarin, and tomato fruits) were 5.76 ± 0.16, 5.35 ± 0.07, and 6.09 ± 0.12 log CFU/fruit. Levels of *L. monocytogenes* cells were reduced after SAEW + FA treatment by 2.93 ± 0.41, 5.35 ± 0.00, and 3.31 ± 0.86 log, respectively (Figure 6B). For apple and tomato, CaO-FA + SAEW treatment produced a more significant (*p* < 0.05) reduction than SAEW + FA, resulting in 4.06 ± 0.07 and 4.68 ± 0.32 log reductions, respectively. 

## 4. Conclusions

The results of this study indicate that the combination of CaO, FA, and SAEW treatments for the disinfection of fresh apple, mandarin, and tomato is a promising approach to reducing microbial risk. The combined treatment can reduce the initial microbial load and the survival and/or growth of natural microbes significantly. In addition to its antibacterial effect, the treatments can help maintain the sensory quality of the treated products during storage. CaO can enhance the bactericidal effect of FA and SAEW, but the contribution is limited. Considering the advantages of the hurdle technology developed by this study, SAEW and FA have great potential for disinfecting microbes of fresh produce, such as fruits and vegetables. Application of calcium to apple, mandarin, and tomato can reduce the number of the pathogens such as *E. coli* O157:H7 and *L. monocytogenes*. As pointed out previously, the new combination treatment of CaO, FA, and SAEW has great advantages in reducing the negative impact experienced by the fresh produce industry and ensuring the quality and safety of fresh fruits.

## Figures and Tables

**Figure 1 foods-08-00497-f001:**
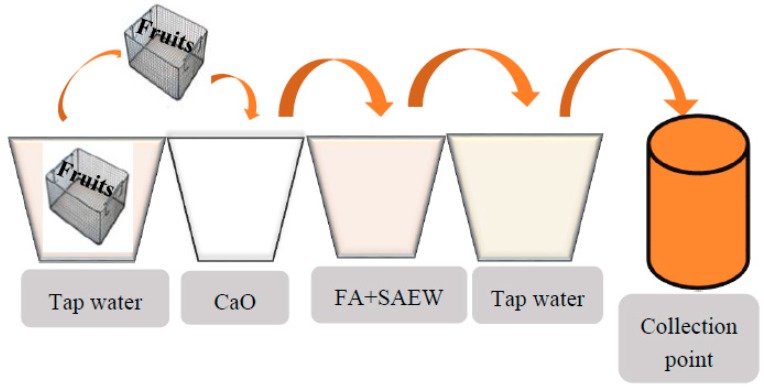
Industrial-scale sequential treatment line developed at Nette Company. Each kind of fruit was separated into three groups. The first group of fresh samples (2 kg) was washed in a treatment tank containing 180 L of slightly acidic electrolyzed water (SAEW) combined with 0.5% (w/v) fumaric acid (FA) (FA+SAEW) for 3 min at room temperature (RT, 23 °C) and then rinsed with tap water for 1 min. For the second group, the fresh samples (2 kg) were dipped in 180 L of CaO for 3 min, and followed with the first round treatment. In the third group, samples were washed in 180 L of tap water for 5 min and this group served as control. After the treatments, autoclaved towel tissues were used to wipe off excess chemical solution or tap water from the surface of treated samples. The fruits were individually packed in sterile stomacher bag and transported to the laboratory under low temperature for microbiological analysis.

**Figure 2 foods-08-00497-f002:**
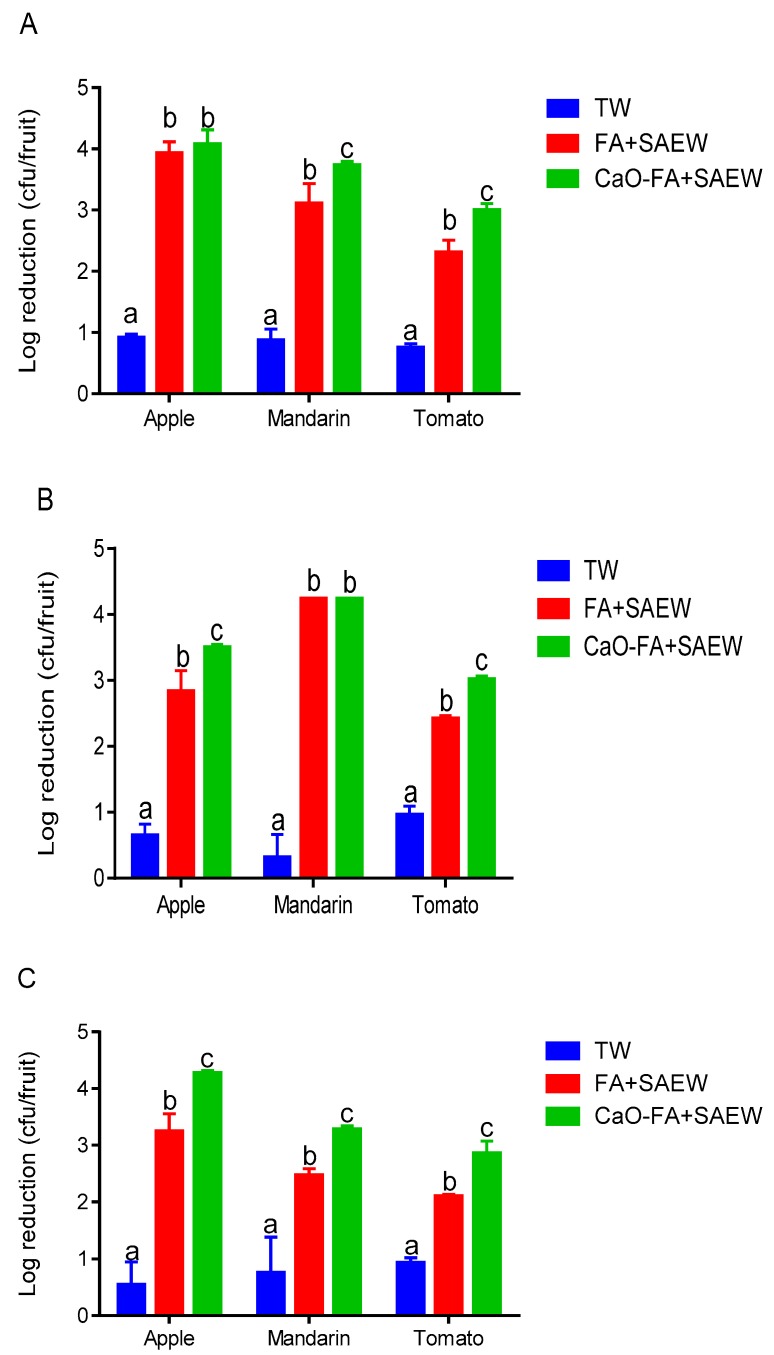
Natural microbiota reduction on different fresh fruits washed with tap water (TW), fumaric acid combined with SAEW (FA+SAEW), and calcium oxide followed by combination (CaO-FA+SAEW). (**A**) Total aerobic bacteria; (**B**) total coliforms; (**C**) yeast and mold. Vertical bars represent standard error of the mean (*n* = 3), different letters in the same group indicate a significant (*p* < 0.05) treatment effect.

**Figure 3 foods-08-00497-f003:**
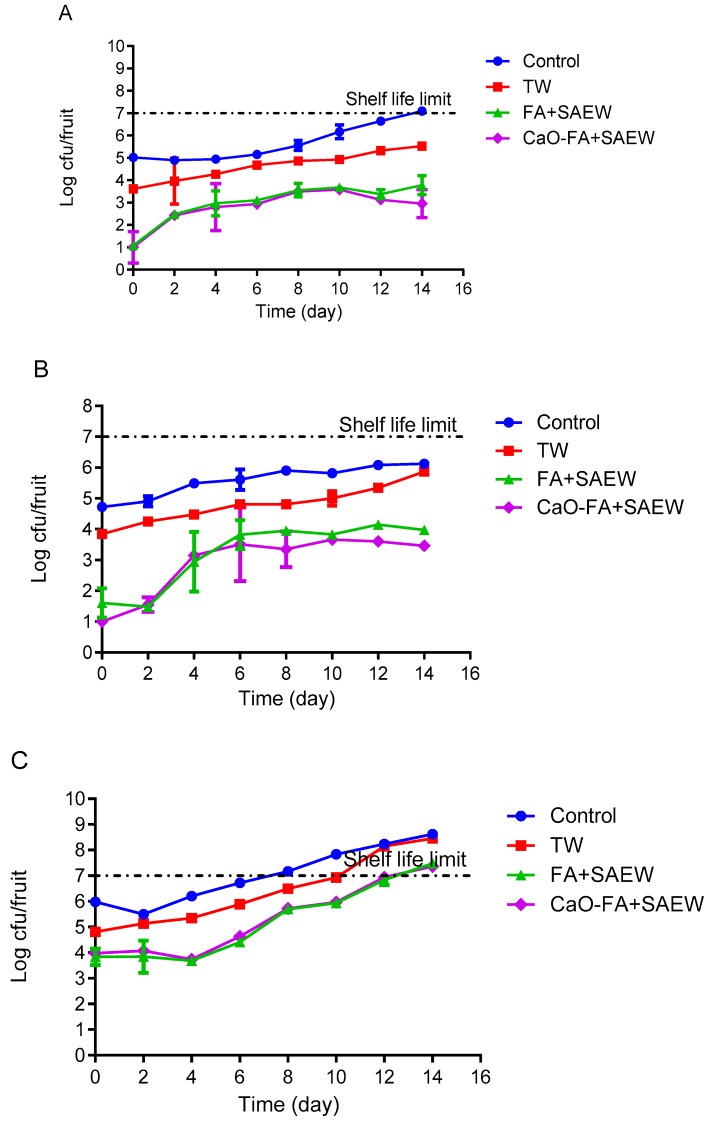
The growth of total aerobic bacteria on fruits during storage at 4 °C after different treatments. TW: Tap water; FA+SAEW: Combination of FA and SAEW treatment; CaO-FA+SAEW: Calcium oxide washing followed by combination of FA+SAEW treatment. Red line represents the end of shelf-life. (**A**) Apple; (**B**) mandarin; (**C**) tomato. Vertical bars represent standard error of the mean (*n* = 3).

**Figure 4 foods-08-00497-f004:**
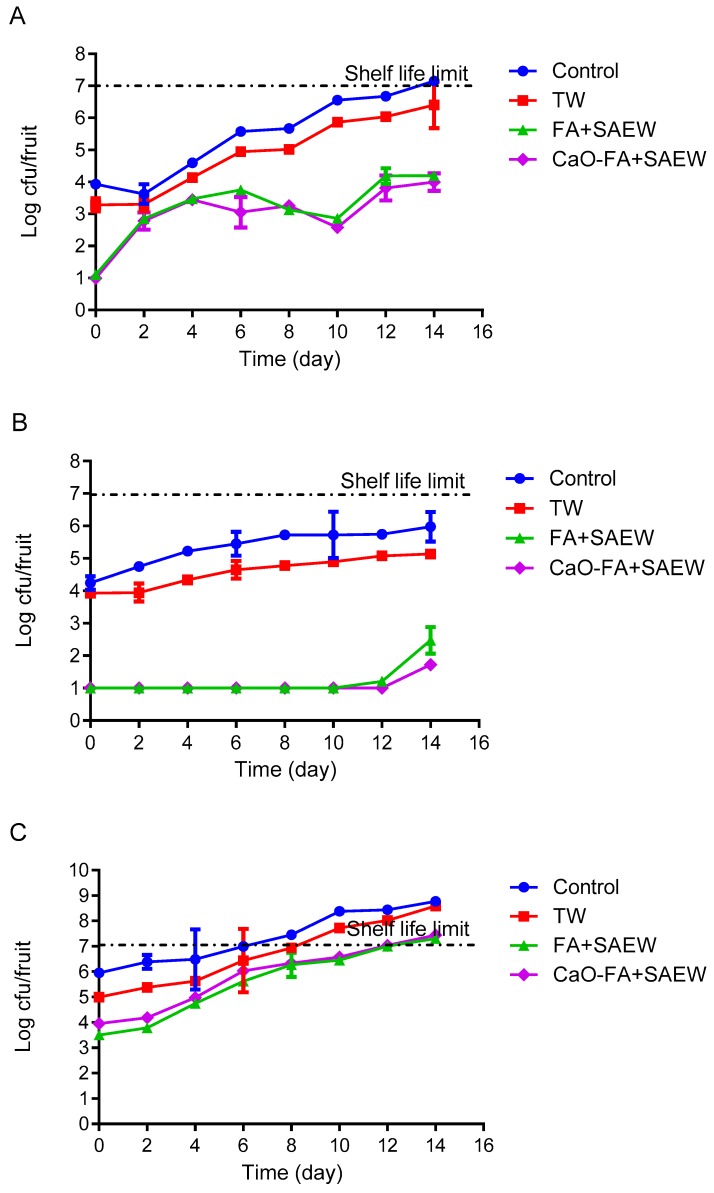
The growth of total coliforms on fruits during storage at 4 °C after different treatments. TW: Tap water; FA+SAEW: Combination of FA and SAEW treatment; CaO-FA+SAEW: Calcium oxide washing followed by combination of FA+SAEW treatment. Red line represents the end of shelf-life. (**A**) Apple; (**B**) mandarin; (**C**) tomato. Vertical bars represent standard error of the mean (*n* = 3).

**Figure 5 foods-08-00497-f005:**
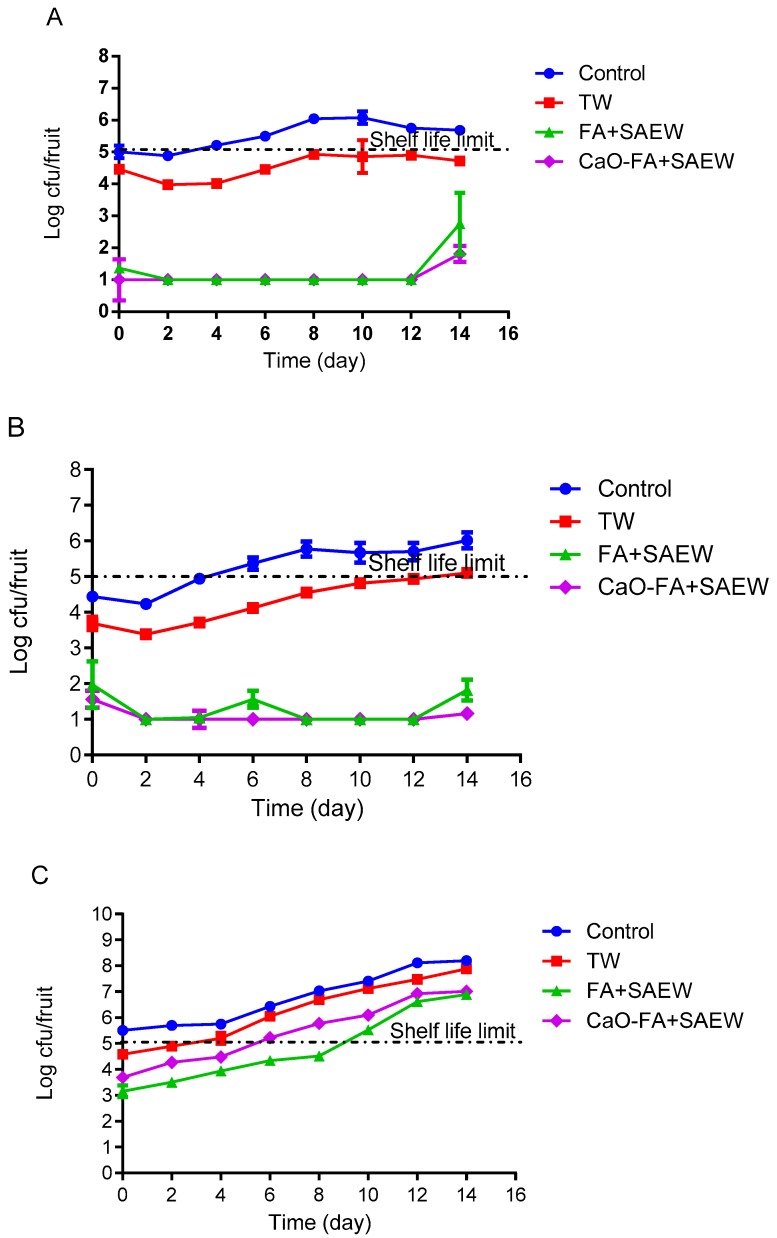
The growth of yeast and mold on fruits during storage at 4 °C after different treatments. TW: Tap water; FA+SAEW: Combination of FA and SAEW treatment; CaO-FA+SAEW: Calcium oxide washing followed by combination of FA+SAEW treatment. Red line represents the end of shelf-life. (**A**) Apple; (**B**) mandarin; (**C**) tomato. Vertical bars represent standard error of the mean (*n* = 3).

**Figure 6 foods-08-00497-f006:**
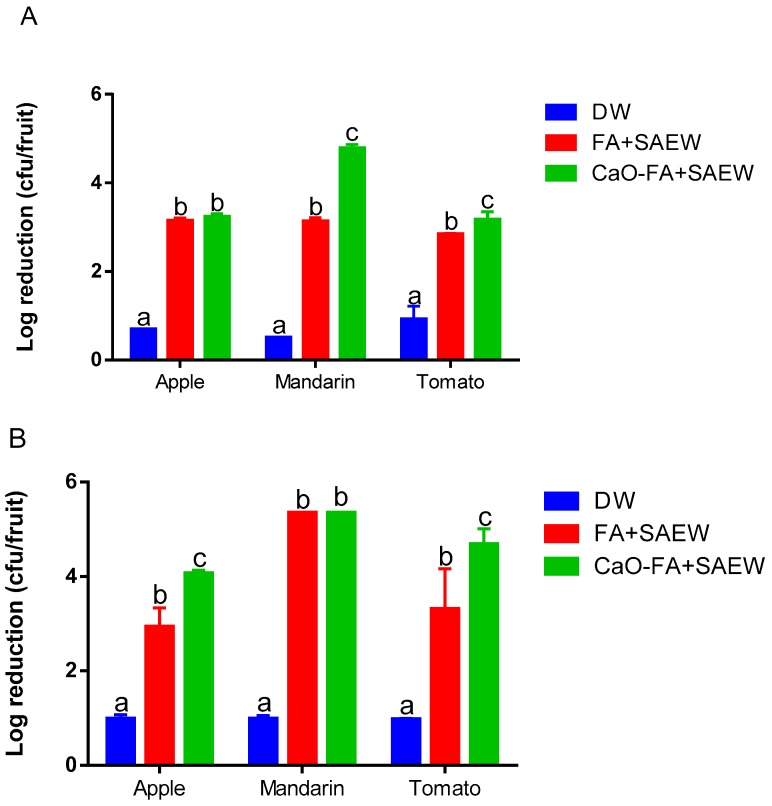
Pathogen reduction on different fruits washed with Tap water (TW), fumaric acid combined with SAEW (FA+SAEW), and calcium oxide followed by combined FA with SAEW treatment (CaO-FA+SAEW). (**A**) *E. coli* O157:H7; (**B**) *L. monocytogenes*. Vertical bars represent standard error of the mean (*n* = 3), different letters in the same group indicate a significant (*p* < 0.05) treatment effect.

**Table 1 foods-08-00497-t001:** Sensory quality changes of fresh apple during storage at 4 °C.

Parameter	Treatment	0 Day	2 Day	4 Day	6 Day	8 Day	10 Day	12 Day	14 Day
Appearance	Unwashed	9.00 ± 0.00 dA	9.00 ± 0.00 dA	9.00 ± 0.00 dA	8.34 ± 0.47 cdA	8.00 ± 0.00 cA	6.67 ± 0.48 bA	6.00 ± 0.00 bA	4.67 ± 0.48 aA
	Tap water	9.00 ± 0.00 dA	9.00 ± 0.00 dA	9.00 ± 0.00 dA	9.00 ± 0.00 dA	8.34 ± 0.048 cdA	7.34 ± 0.48 bcA	6.34 ± 0.48 bA	4.67 ± 0.48 aA
	FA+SAEW	9.00 ± 0.00 bA	9.00 ± 0.00 bA	9.00 ± 0.00 bA	9.00 ± 0.00 bA	8.34 ± 0.48 bA	7.34 ± 0.95 bA	6.67 ± 0.48 aAB	5.67 ± 0.48 aAB
	C-F+S	9.00 ± 0.00 bA	9.00 ± 0.00 bA	9.00 ± 0.00 bA	9.00 ± 0.00 bA	8.34 ± 0.48 bA	7.47 ± 0.48 bA	7.00 ± 0.00 abB	6.67 ± 0.95 aB
Color	Unwashed	9.00 ± 0.00 cA	9.00 ± 0.00 cA	9.00 ± 0.00 cA	8.67 ± 0.48 cA	8.34 ± 0.48 cA	7.67 ± 0.95 bcA	6.67 ± 0.48 bA	4.00 ± 0.00 aA
	Tap water	9.00 ± 0.00 bA	9.00 ± 0.00 bA	9.00 ± 0.00 bA	9.00 ± 0.00 bA	8.00 ± 0.82 abA	7.34 ± 0.48 aA	7.00 ± 0.82 aA	4.67 ± 0.48 aAB
	FA+SAEW	9.00 ± 0.00 bA	9.00 ± 0.00 bA	9.00 ± 0.00 bA	9.00 ± 0.00 bA	8.34 ± 0.48 bA	8.34 ± 0.48 bA	6.34 ± 0.48 aA	5.34 ± 0.48 aB
	C-F+S	9.00 ± 0.00 cA	9.00 ± 0.00 cA	9.00 ± 0.00 cA	9.00 ± 0.00 cA	8.67 ± 0.48 cA	7.67 ± 0.48 bA	7.00 ± 0.00 bA	5.00 ± 0.00 aB
Off-odor	Unwashed	9.00 ± 0.00 aA	9.00 ± 0.00 aA	9.00 ± 0.00 aA	8.34 ± 0.48 aA	8.66 ± 0.48 aA	8.00 ± 0.82 aA	7.67 ± 0.48 aA	7.67 ± 0.48 aA
	Tap water	9.00 ± 0.00 bA	9.00 ± 0.00 bA	9.00 ± 0.00 bA	9.00 ± 0.00 abA	9.00 ± 0.00 abA	9.00 ± 0.00 bA	8.34 ± 0.48 abAB	8.00 ± 0.00 aA
	FA+SAEW	9.00 ± 0.00 bA	9.00 ± 0.00 bA	9.00 ± 0.00 bA	9.00 ± 0.00 bA	9.00 ± 0.00 bA	9.00 ± 0.00 aA	9.00 ± 0.00 aB	8.00 ± 0.00 aA
	C-F+S	9.00 ± 0.00 bA	9.00 ± 0.00 bA	9.00 ± 0.00 bA	9.00 ± 0.00 bA	9.00 ± 0.00 bA	9.00 ± 0.00 abA	9.00 ± 0.00 aB	8.00 ± 0.00 aA
Overall	Unwashed	9.00 ± 0.00 cA	9.00 ± 0.00 cA	9.00 ± 0.00 cA	8.67 ± 0.48 cA	8.34 ± 0.48 cA	7.00 ± 0.82 bA	6.00 ± 0.00 abA	5.00 ± 0.00 aA
	Tap water	9.00 ± 0.00 dA	9.00 ± 0.00 dA	9.00 ± 0.00 dA	9.00 ± 0.00 dA	8.00 ± 0.00 cAB	7.34 ± 0.48 cA	6.34 ± 0.48 bA	5.00 ± 0.00 aA
	FA+SAEW	9.00 ± 0.00 bA	9.00 ± 0.00 bA	9.00 ± 0.00 bA	9.00 ± 0.00 bA	9.00 ± 0.00 bB	9.00 ± 0.00 bB	6.67 ± 0.48 aA	5.34 ± 0.48 aAB
	C-F+S	9.00 ± 0.00 bA	9.00 ± 0.00 bA	9.00 ± 0.00 bA	9.00 ± 0.00 bA	9.00 ± 0.00 bB	9.00 ± 0.00 bB	7.34 ± 0.48 aA	6.67 ± 0.48 aC

Different letters in same group (lowercase) or same row (capital) indicate a significant (*p* < 0.05) treatment effect. NaOCl: Sodium hypochlorite, FA+SAEW: Fumaric acid combined with SAEW, and C-F+S: Calcium oxide washing followed by combination of FA+SAEW (CaO-FA+SAEW). Values are mean ± SD of 3 replications, different letters in the same group indicate a significant (*p* < 0.05) treatment effect.

**Table 2 foods-08-00497-t002:** Sensory quality changes of fresh mandarin during storage at 4 °C.

Parameter	Treatment	0 Day	2 Day	4 Day	6 Day	8 Day	10 Day	12 Day	14 Day
Appearance	Unwashed	9.00 ± 0.00 eA	9.00 ± 0.00 eA	8.00 ± 0.00 dAB	8.00 ± 0.00 dA	8.00 ± 0.00 dA	6.67 ± 0.48 cA	5.67 ± 0.48 bA	4.34 ± 0.48 aA
	Tap water	9.00 ± 0.00 dA	9.00 ± 0.00 dA	8.00 ± 0.00 cA	8.00 ± 0.00 cA	8.00 ± 0.00 cA	7.34 ± 0.48 cAB	5.67 ± 0.48 bA	4.67 ± 0.48 aA
	FA+SAEW	9.00 ± 0.00 cA	9.00 ± 0.00 cA	9.00 ± 0.00 bB	8.00 ± 0.00 bA	8.00 ± 0.00 bA	7.67 ± 0.48 bAB	6.67 ± 0.48 aA	6.00 ± 0.00 aAB
	C-F+S	9.00 ± 0.00 bA	9.00 ± 0.00 bA	9.00 ± 0.00 bB	8.00 ± 0.00 abA	8.00 ± 0.00 abA	8.67 ± 0.48 bB	6.67 ± 0.48 aA	6.67 ± 0.95 aB
Color	Unwashed	9.00 ± 0.00 eA	9.00 ± 0.00 eA	8.00 ± 0.00 dAB	8.00 ± 0.00 dAB	8.00 ± 0.00 dAB	6.34 ± 0.48 cA	5.00 ± 0.00 bA	4.00 ± 0.00 aA
	Tap water	9.00 ± 0.00 dA	9.00 ± 0.00 dA	8.00 ± 0.00 cA	8.00 ± 0.00 cA	8.00 ± 0.00 cA	6.67 ± 0.48 bA	6.34 ± 0.48 bB	4.34 ± 0.48 aAB
	FA+SAEW	9.00 ± 0.00 cA	9.00 ± 0.00 cA	9.00 ± 0.00 cB	9.00 ± 0.00 cB	9.00 ± 0.00 cB	6.67 ± 0.48 bA	6.34 ± 0.48 bB	5.00 ± 0.00 aB
	C-F+S	9.00 ± 0.00 cA	9.00 ± 0.00 cA	9.00 ± 0.00 cB	9.00 ± 0.00 cB	9.00 ± 0.00 cB	7.34 ± 0.48 bA	7.00 ± 0.00 bB	5.00 ± 0.00 aB
Off-odor	Unwashed	9.00 ± 0.00 cA	9.00 ± 0.00 cA	9.00 ± 0.00 cA	8.00 ± 0.00 bAB	8.00 ± 0.00 bAB	7.67 ± 0.48 bA	8.00 ± 0.00 bA	7.00 ± 0.00 aA
	Tap water	9.00 ± 0.00 bA	9.00 ± 0.00 bA	9.00 ± 0.00 bA	8.00 ± 0.00 aA	8.00 ± 0.00 aA	7.67 ± 0.48 aA	7.34 ± 0.48 aA	8.00 ± 0.00 aA
	FA+SAEW	9.00 ± 0.00 bA	9.00 ± 0.00 bA	9.00 ± 0.00 bA	9.00 ± 0.00 bB	9.00 ± 0.00 bB	8.00 ± 0.00 aA	8.00 ± 0.00 aA	7.67 ± 0.48 aA
	C-F+S	9.00 ± 0.00 cA	9.00 ± 0.00 cA	9.00 ± 0.00 cA	9.00 ± 0.00 cB	9.00 ± 0.00 cB	8.34 ± 0.48 bA	8.00 ± 0.00 bA	7.67 ± 0.48 aA
Overall	Unwashed	9.00 ± 0.00 eA	9.00 ± 0.00 eA	9.00 ± 0.00 eA	8.00 ± 0.00 dAB	8.00 ± 0.00 dAB	6.00 ± 0.00 cA	5.00 ± 0.00 bA	5.00 ± 0.00 aA
	Tap water	9.00 ± 0.00 dA	9.00 ± 0.00 dA	9.00 ± 0.00 dA	8.00 ± 0.00 cdA	8.00 ± 0.00 cA	7.34 ± 0.48 cA	6.00 ± 0.00 bAB	5.67 ± 0.48 aB
	FA+SAEW	9.00 ± 0.00 dA	9.00 ± 0.00 dA	9.00 ± 0.00 dA	9.00 ± 0.00 dB	9.00 ± 0.00 dB	8.00 ± 0.00 cA	6.67 ± 0.48 bBC	6.00 ± 0.00 aBC
	C-F+S	9.00 ± 0.00 dA	9.00 ± 0.00 dA	9.00 ± 0.00 dA	9.00 ± 0.00 dB	9.00 ± 0.00 dB	8.00 ± 0.00 cA	7.34 ± 0.48 bC	6.00 ± 0.00 aC

Different letters in same group (lowercase) or same row (capital) indicate a significant (*p* < 0.05) treatment effect. NaOCl: Sodium hypochlorite, FA+SAEW: Fumaric acid combined with SAEW, and C-F+S: Calcium oxide washing followed by combination of FA+SAEW (CaO-FA+SAEW). Values are mean ± SD of 3 replications, different letters in the same group indicate a significant (*p* < 0.05) treatment effect.

**Table 3 foods-08-00497-t003:** Sensory quality changes of fresh tomato during storage at 4 °C.

Parameter	Treatment	0 Day	2 Day	4 Day	6 Day	8 Day	10 Day	12 Day	14 Day
Appearance	Unwashed	9.00 ± 0.00aC	8.33 ± 0.47aC	7.33 ± 0.47aB	7.33 ± 0.47aB	7.33 ± 0.47a	7.33 ± 0.47aB	6.67 ± 0.47aB	3.00 ± 1.41aA
	Tap water	9.00 ± 0.00aC	9.00 ± 0.00aC	8.00 ± 0.00aBC	8.00 ± 0.00aBC	8.00 ± 0.00aBC	8.00 ± 0.00aBC	7.00 ± 0.00abB	5.67 ± 0.47bA
	NaOCl	9.00 ± 0.00aC	9.00 ± 0.00aC	8.00 ± 0.00aBC	8.00 ± 0.00aBC	8.00 ± 0.00aBC	8.00 ± 0.00aBC	7.00 ± 0.00abB	5.67 ± 0.47bA
	FA+SAEW	9.00 ± 0.00aB	9.00 ± 0.00aB	8.00 ± 0.00aB	8.00 ± 0.00aB	8.00 ± 0.00aB	8.00 ± 0.00aB	8.00 ± 0.00bB	6.33 ± 0.94bcA
	C-F+S	9.00 ± 0.00aB	9.00 ± 0.00aB	8.00 ± 0.00aB	8.00 ± 0.00aB	8.00 ± 0.00aB	8.00 ± 0.00aB	8.00 ± 0.00bB	6.67 ± 0.47cA
Color	Unwashed	9.00 ± 0.00aD	8.33 ± 0.47aD	7.33 ± 0.47aC	6.67 ± 0.47aBC	6.33 ± 0.47aBC	6.33 ± 0.47aBC	6.00 ± 0.00aB	3.33 ± 1.25aA
	Tap water	9.00 ± 0.00aD	9.00 ± 0.00aD	8.00 ± 0.00aC	8.00 ± 0.00bC	7.33 ± 0.47bBC	7.33 ± 0.47bBC	7.00 ± 0.00bB	5.33 ± 0.94bA
	NaOCl	9.00 ± 0.00aD	9.00 ± 0.00aD	8.00 ± 0.00aC	8.00 ± 0.00bC	7.33 ± 0.47bBC	7.33 ± 0.47bBC	7.00 ± 0.00bB	5.33 ± 0.94bA
	FA+SAEW	9.00 ± 0.00aB	9.00 ± 0.00aB	8.00 ± 0.00aB	8.00 ± 0.00bB	8.00 ± 0.00bB	8.00 ± 0.00bB	8.00 ± 0.00cB	6.67 ± 0.47cA
	C-F+S	9.00 ± 0.00aC	9.00 ± 0.00aC	8.00 ± 0.00aAB	8.00 ± 0.00bAB	8.00 ± 0.00bAB	8.00 ± 0.00bAB	7.67 ± 0.47bcA	7.00 ± 0.00dA
Off-odor	Unwashed	9.00 ± 0.00aC	8.33 ± 0.47aC	8.33 ± 0.47aC	8.33 ± 0.47aC	7.00 ± 0.00aB	7.00 ± 0.00aB	6.33 ± 0.47aB	4.00 ± 0.82aA
	Tap water	9.00 ± 0.00aC	9.00 ± 0.00aC	9.00 ± 0.00aC	9.00 ± 0.00aC	8.00 ± 0.00bBC	8.00 ± 0.00bBC	7.67 ± 0.47bB	5.67 ± 0.47bA
	NaOCl	9.00 ± 0.00aBC	9.00 ± 0.00aBC	9.00 ± 0.00aBC	9.00 ± 0.00aBC	8.00 ± 0.00bBC	8.00 ± 0.00bBC	7.67 ± 0.47bB	5.67 ± 0.47bA
	FA+SAEW	9.00 ± 0.00aC	9.00 ± 0.00aC	9.00 ± 0.00aC	9.00 ± 0.00aC	9.00 ± 0.00cC	9.00 ± 0.00cC	7.67 ± 0.47bB	6.33 ± 0.94bcA
	C-F+S	9.00 ± 0.00aB	9.00 ± 0.00aB	9.00 ± 0.00aB	9.00 ± 0.00aB	9.00 ± 0.00cB	9.00 ± 0.00cB	8.00 ± 0.00bB	6.67 ± 0.47cA
Overall	Unwashed	9.00 ± 0.00aE	8.00 ± 0.00aD	7.67 ± 0.47aCD	7.33 ± 0.47aCD	7.00 ± 0.00aC	7.00 ± 0.00aC	6.00 ± 0.82aB	4.00 ± 0.82aA
	Tap water	9.00 ± 0.00aD	9.00 ± 0.00Da	8.67 ± 0.47bCD	8.00 ± 0.00abC	7.67 ± 0.47abBC	7.67 ± 0.47abBC	7.00 ± 0.00bB	5.33 ± 0.94bA
	NaOCl	9.00 ± 0.00aD	9.00 ± 0.00aD	8.67 ± 0.47bCD	8.00 ± 0.00abC	7.67 ± 0.47abBC	7.67 ± 0.47abBC	7.00 ± 0.00bB	5.33 ± 0.94bA
	FA+SAEW	9.00 ± 0.00aB	9.00 ± 0.00aB	9.00 ± 0.00bB	9.00 ± 0.00bB	8.00 ± 0.00bB	8.00 ± 0.00bB	8.00 ± 0.00cB	6.67 ± 0.47cA
	C-F+S	9.00 ± 0.00aC	9.00 ± 0.00aC	9.00 ± 0.00bC	9.00 ± 0.00bC	8.33 ± 0.47bBC	8.33 ± 0.47bBC	8.00 ± 0.00cB	7.00 ± 0.00cA

Different letters in same group (lowercase) or same row (capital) indicate a significant (*p* < 0.05) treatment effect. NaOCl: Sodium hypochlorite, FA+SAEW: Fumaric acid combined with SAEW, and C-F+S: Calcium oxide washing followed by combination of FA + SAEW (CaO-FA+SAEW). Values are mean ± SD of 3 replications, different letters in the same group indicate a significant (*p* < 0.05) treatment effect.

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
