# Peer review of "Disinfection Efficacy of Slightly Acidic Electrolyzed Water Combined with Chemical Treatments on Fresh Fruits at the Industrial Scale"

_foods, 2019, doi:10.3390/foods8100497_

Round 1

Reviewer 1 Report

In the present manuscript, the author observed inactivation efficiency of slightly electrolysed water in combination with Fumaric acid and calcium oxide against pathogenic and natural microflora population in fresh fruits. Recently many similar results have been published from different journals even though the fresh fruits/produce selection may be slightly different. Overall in general, disinfection efficacy of combination of SAEW in combination with Fumaric acid and calcium oxide for microbial quality for fresh produce has been demonstrated before; therefore this study has limited contribution.

Author Response

Comments and Suggestions for Authors

In the present manuscript, the author observed inactivation efficiency of slightly electrolyzed water in combination with Fumaric acid and calcium oxide against pathogenic and natural microflora population in fresh fruits. Recently many similar results have been published from different journals even though the fresh fruits/produce selection may be slightly different. Overall in general, disinfection efficacy of combination of SAEW in combination with Fumaric acid and calcium oxide for microbial quality for fresh produce has been demonstrated before; therefore this study has limited contribution.

Response: The present study was performed to validate the combined treatment (CaO-FA+SAEW) in industrial scale. Most studies regarding the effect of combined treatment, especially for FA and SAEW have been performed in the Laboratory. However, significant difference may appear when the experiment is carried out under industrial scale. The present study demonstrated the effectiveness of this hurdle technology in industrial scale.

Reviewer 2 Report

This manuscript describes the utilization of acidic electrolyzed water combined with fumaric acid and calcium oxide on fruits in the disinfection of fruits as alternatives to sodium hypochlorite in the industry. The manuscript needs an upgrading in the Introduction, Discussion of the results and in the Conclusions.

Introduction

The “Introduction” should be improved with information about the utilization of fumaric acid and calcium oxide in the fruit disinfection including advantages and disadvantages of the utilization of those chemicals.

In line 46 add some examples of decontamination using acidic electrolyzed water applied to fresh-cut fruits (apple, pear, mango, etc).

Line 50- The application of electrolyzed water as a decontaminant of fruits is not novel.  Change the phrases and cite previous studies done in fruits (whole and fresh-cut).

Line 58 - The citation is not correctly introduced in the text

Line 63- on enhancing SAEW

Line 64- correct the citation  

Line 72- that combine

Line 74- in an industry that processes fresh fruits.

Material and methods

Line 81- experiment. Each polyethylene …, three batches per fruit were used … .

Line 85- Sigma

Line 86 – What do the authors mean when using the expression “self-developed”.

Line 93- The SAEW were collected … (4 °C) and used rapidly in the …

Line 119- mold petri film? This petri film is not used to count coliforms. Please correct.

Line 136- Is the word “target” needed?

Line 146- I do not understand how the pathogenic contamination mixture was done. Make the phrase clearer.

Line 151- What do the authors mean with the expression: “is kept operation”?

Line 152- Why do the fruits were subjected to ultraviolet? What kind of ultraviolet?

Line 162-The authors have to describe the control treatment.

Line 172- Were the bags closed or opened?

Line 177- The Compertz model was not used. What are the values of the model parameters? I do not see them anywhere in the manuscript. There are no results of growth kinetics. So, erase the phrase.

Line 193-197- phrases need reformulation.

Results and Discussion

 The authors should improve the discussion of the results comparing the microbial reductions they obtained with the microbial reductions described in other articles, when electrolyzed water was used to decontaminate fruits. The authors should discuss why the microbial reductions they obtained in the fruits tested are higher than the ones obtained in minimally processed fruits.

Line 204- FA+SAWE. The results may be explained by the fact that CaO has a pH of about 12.5 which can be associated to antimicrobial activity by destroying the outer membrane and interfering in the microbial enzymatic activity.

Line 224- comparing to

Line 246- I do not understand the phrase

Line 276- resulted in a lower …

Fig.4 and 5- What was the detection limit for coliforms and yeasts and molds? 0 ufc? I think not. The authors should change the graphs and include the detection limit which is not  0 ufc.

Line 298-  … after 12 days …

Conclusions

The conclusions ought to be exclusively based on the results obtained from the study described in the manuscript.

Lines 351- 354- eliminate from “Due to ….. human health” (it is not a conclusion from the study described)

Lines 357- can induce or can reduce ?

Line 358- eliminate the phrase from “and calcium … and tomato”.

References

The authors should verify the references as some mistakes are found:

Some journal names are abbreviated and others not.

Pay attention to the words in italics.

Author Response

Comments and Suggestions for Authors

This manuscript describes the utilization of acidic electrolyzed water combined with fumaric acid and calcium oxide on fruits in the disinfection of fruits as alternatives to sodium hypochlorite in the industry. The manuscript needs an upgrading in the Introduction, Discussion of the results and in the Conclusions.

Response: Thank you so much for your time and suggestions. Indeed your comments are very valuable for us.

According to your suggestions we have modify some parts of the introduction, discussion of the results and the conclusions to more clarify the whole theme of the paper (Lines 51 to 56; 77 to 85).

Introduction

The “Introduction” should be improved with information about the utilization of fumaric acid and calcium oxide in the fruit disinfection including advantages and disadvantages of the utilization of those chemicals.

Response: We have included the utilization fumaric acid and calcium oxide in the fruit disinfection which will explain the whole theme of the paper as you suggested (Line 77 to 85).

In line 46 add some examples of decontamination using acidic electrolyzed water applied to fresh-cut fruits (apple, pear, mango, etc).

Response: We added some examples related to decontamination using AEW (Lines 51 to 56)

Line 50- The application of electrolyzed water as a decontaminant of fruits is not novel.  Change the phrases and cite previous studies done in fruits (whole and fresh-cut).

Response: The word “novel” was replaced by “environmentally friendly”

Line 58 - The citation is not correctly introduced in the text

Response: We corrected it

Line 63- on enhancing SAEW

Response: We corrected it

Line 64- correct the citation

Response: We corrected it 

Line 72- that combine

Response: We corrected it 

Line 74- in an industry that processes fresh fruits.

Response: We corrected it 

Material and methods

Line 81- experiment. Each polyethylene …, three batches per fruit were used.

Response: We corrected it 

Line 85- Sigma

Response: We corrected it 

Line 86 – What do the authors mean when using the expression “self-developed”.

Response: (Self- developed, means we buy separate parts, assemble by ourselves according the principal of EW generation)

Line 93- The SAEW were collected … (4 °C) and used rapidly in the …

Response: We corrected it 

Line 119- mold petri film? This petri film is not used to count coliforms. Please correct.

Response: We corrected it (Lines 132 to 134)

Line 136- Is the word “target” needed?

Response: we removed it

Line 146- I do not understand how the pathogenic contamination mixture was done. Make the phrase clearer.

Response: We revised it (Line 159 to 161)

Line 151- What do the authors mean with the expression: “is kept operation”?

Response: We revised it (Line 164 to 165)

Line 152- Why do the fruits were subjected to ultraviolet? What kind of ultraviolet?

Response: the fruits were exposed to ultraviolet light to reduce the natural microflora before inoculation.

Line 162-The authors have to describe the control treatment.

Response: We firstly washed 3 min in the first tank containing TW followed by 2 min in second tank to have the same washing time compare to the treatment.

 Line 172- Were the bags closed or opened?

Response: the fruits were sealed into the bag before storage

Line 177- The Compertz model was not used. What are the values of the model parameters? I do not see them anywhere in the manuscript. There are no results of growth kinetics. So, erase the phrase.

Response: we deleted it

Line 193-197- phrases need reformulation.

Response: We reformulated the phrases (Lines 208 to 212).

Results and Discussion

 The authors should improve the discussion of the results comparing the microbial reductions they obtained with the microbial reductions described in other articles, when electrolyzed water was used to decontaminate fruits. The authors should discuss why the microbial reductions they obtained in the fruits tested are higher than the ones obtained in minimally processed fruits.

Line 204- FA+SAWE. The results may be explained by the fact that CaO has a pH of about 12.5 which can be associated to antimicrobial activity by destroying the outer membrane and interfering in the microbial enzymatic activity.

Response: we revised as you suggested (Lines 219 to 221)

Line 224- comparing to

Response: we revised as you suggested

Line 246- I do not understand the phrase

Response: we revised this phrase (Line 262).

Line 276- resulted in a lower …

Response: we revised as you suggested

Fig.4 and 5- What was the detection limit for coliforms and yeasts and molds? 0 ufc? I think not. The authors should change the graphs and include the detection limit which is not 0 ufc.

Response: The detection limit of the used method was 1.0 cfu/g. under the detection limit, the bacterial count was considered as zero, and therefore, we represented, in curves, all the non-detected count as zero.   

Line 298- … after 12 days …

Response: we revised it as you suggested

Conclusions

The conclusions ought to be exclusively based on the results obtained from the study described in the manuscript.

Lines 351- 354- eliminate from “Due to ….. Human health” (it is not a conclusion from the study described)

Response: we removed it as you suggested

Lines 357- can induce or can reduce?

Response: we revised it as “can reduce”

Line 358- eliminate the phrase from “and calcium … and tomato”.

Response: We eliminated it as you suggested 

References

The authors should verify the references as some mistakes are found:

Some journal names are abbreviated and others not.

Pay attention to the words in italics.

Response: we carefully checked and corrected the references

Reviewer 3 Report

Disinfection efficacy of slightly acidic electrolyzed water combined with chemical treatments on fresh fruits

 I write you in regard to manuscript # foods-583603 entitled "Disinfection efficacy of slightly acidic electrolyzed water combined with chemical treatments on fresh fruits" which you submitted to the foods.

I confuse about the novelty of this manuscript because authors already published a similar paper with a similar treatment in almost similar fruits (apple and tomato) except mandarin and the same storage temperature. This manuscript was Tango, C.N.; Khan, I.; Ngnitcho Kounkeu, P.F.; Momna, R.; Hussain, M.S.; Oh, D.H. Slightly acidic electrolyzed water combined with chemical and physical treatments to decontaminate bacteria on fresh fruits. Food microbiology 2017, 67, 97-105.’ http://dx.doi.org/10.1016/j.fm.2017.06.007

In the present manuscript (foods-583603), authors were stored treated fruits (apple, mandarin, and tomato) for 14 days. They should know the shelf life of an apple, mandarin, and tomato without treatment. These treatments for 14 days are not meaningful.

In fruits type of experiment, authors should confirm the maturity stage because maturity stage influences the storage condition and measured parameters.

Author Response

Comment:

I confuse about the novelty of this manuscript because authors already published a similar paper with a similar treatment in almost similar fruits (apple and tomato) except mandarin and the same storage temperature. This manuscript was Tango, C.N.; Khan, I.; Ngnitcho Kounkeu, P.F.; Momna, R.; Hussain, M.S.; Oh, D.H. Slightly acidic electrolyzed water combined with chemical and physical treatments to decontaminate bacteria on fresh fruits. Food microbiology 2017, 67, 97-105.’

Response: The present study was performed to validate the combined treatment (CaO-FA+SAEW) in industrial scale. Most studies regarding the effect of combined treatment, especially for FA and SAEW have been performed in the Laboratory. However, significant difference may appear when the experiment is carried out under industrial scale. The present study demonstrated the effectiveness of this hurdle technology in industrial scale.

In the present manuscript (foods-583603), authors were stored treated fruits (apple, mandarin, and tomato) for 14 days. They should know the shelf life of an apple, mandarin, and tomato without treatment. These treatments for 14 days are not meaningful.

Response: We thank the reviewer for his important comment. However, in the microbial decontamination study, the shelf life of fruits and vegetables is usually evaluated during 14-21 days of storage at 4 °C after treatment. Please refer to.

Gomez-Lopez et al. Shelf-life of minimally processed lettuce and cabbage treated with gaseous chlorine dioxide and cysteine (International Journal of Food Microbiology, 2008).

Gómez-López et al. Shelf-life extension of minimally processed carrots by gaseous chlorine dioxide (International Journal of Food Microbiology, 2007).

Debevere, J., 1996. Criteria en praktische methoden voor de bepaling van de houdsbaarheidsdatum in de etikettering. Etikettering, houdsbaarheid en bewaring (voedingsmiddelen en recht 2). Die Keure, Brugge, Belgium, pp. 37–64.

In fruits type of experiment, authors should confirm the maturity stage because maturity stage influences the storage condition and measured parameters.

Response: the fruits were supplied at the commercial maturity. We added the precision in the main text (Line 93).

Round 2

Reviewer 1 Report

The manuscript may be acceptable after some revisions have been made. There are some minor issues that should be checked and some questions that might be worth to address:

Line 52, line 129, line 142, 315: if bacterial species is mentioned before you don’t need to spell it out again.

Same for Listeria monocytogenes in line 142 should spelled in full, when it is mentioned for the first time in text.

Line 74: ‘Calcium oxide’ should be lower case

Line 89, 112 and 114: the name of the supplier company is spelled different in each of these lines. Please correct it to appropriate one.

Line 124- 125: Change the punctuation to pull stop after end of the sentence.

Line 127: This statement is not clear, is the sample suspension 10-fold diluted? Mention the dilution medium used.  

Line 129: E. coli should be in italic; similarly line 350 Bacterial name should be in italics.

145: ‘Strains were inoculated onto above selective agar media from the pathogenic bacteria stock that storage at -80°C.’ This sentence doesn’t make sense. Please rephrase it.

Line 168: define abbreviation when first used in the manuscript. Example: define ‘TW’

Line 179: define MBGA? Why selective agar was changed for L. monocytogenes?

Could you add information on selective supplements used?

Line 202-205: Too long sentence, it doesn’t make sense. Please rephrase it.

Line 214: ‘removes’ should be corrected to remove.

Line 320: ‘Danyluk et al (2019)’ it should be Danyluk et al [37]

Similarly, for reference in line 321 and 326. Use same reference style throughout the manuscript.

Line 334: the abbreviation used for ‘Calcium oxide washing followed by combination of FA + SAEW (CaO-FA + SAEW)’ is different in the table and one mentioned below the table. Please correct it. This is applicable for all the tables.

Table 3: Define the abbreviation ‘NaOCl’ in table text.

Line 371: reference title is repeated twice.

Figures: What was the detection limit for microbiological counts? Please mention them somewhere in the manuscript.

Author Response

We are thanks for your time and valuable comments, we have carefully checked all manuscript and tried to improve the writing as you suggested.

The response to the point of reviewers comments are given below:

The manuscript may be acceptable after some revisions have been made. There are some minor issues that should be checked and some questions that might be worth to address:

Q1: Line 52, line 129, and line 142, 315: if bacterial species is mentioned before you don’t need to spell it out again.

Response: We have revised the text and corrected it. (Line 54, 57, 159, 339).

Q2: Same for Listeria monocytogenes in line 142 should spelled in full, when it is mentioned for the first time in text.

Response: We have revised the text and corrected it. (Line 154).

Q3: Line 74: ‘Calcium oxide’ should be lower case

Response: We corrected it. (Line78) 

Q3: Line 89, 112 and 114: the name of the supplier company is spelled different in each of these lines. Please correct it to appropriate one.

Response: We corrected it according to the comment, name of the supplier company is Nette.

Q4: Line 124- 125: Change the punctuation to pull stop after end of the sentence.

Response: We have corrected the punctuation according to the comment.

Q5: Line 127: This statement is not clear, is the sample suspension 10-fold diluted? Mention the dilution medium used.  

Response: We have revised the statement according to the comment. (Line 136)

“1mL of sample suspension was 10-fold diluted with 0.1% buffered peptone water (BPW Difco).”

Q6: Line 129: E. coli should be in italic; similarly line 350 Bacterial name should be in italics.

Response: We have corrected the format of bacterial names (line 159, 376).

Q6: line145: ‘Strains were inoculated onto above selective agar media from the pathogenic bacteria stock that storage at -80°C.’ This sentence doesn’t make sense. Please rephrase it.

Response: we revised this sentence in the text as suggested by reviewer. (Line 155-156).

“Each suspension (0.1 ml) of the stock cultures was plated on the selective agar media individually.”

Q7: Line 168: define abbreviation when first used in the manuscript. Example: define ‘TW’

Response: We corrected it in the text according to the comment. (Line 181, 183)

Tap water (TW)

Q8: Line 179: define MBGA? Why selective agar was changed for L. monocytogenes?

Could you add information on selective supplements used?

Response: we have changed the MBGA to OBMA in the text. (Line 192) 

We have added the information on selective supplements used. (Line 156-159).

“Oxford base medium agar (OBMA, Difco) is the selective agar for L. monocytogenes, we add information on selective supplements used as your kind suggestion.”

Q9: Line 202-205: Too long sentence, it doesn’t make sense. Please rephrase it.

Response: We have revised it according to the comment. (Line 215-216)

“This study was designed for the fruit process industry with the purpose of improving the quality of fruit products, the experiment is carried out under industrial scale.”

Q10: Line 214: ‘removes’ should be corrected to remove.

Response: We corrected the word as your kind reminder.  

Q11: Line 320: ‘Danyluk et al (2019)’ it should be Danyluk et al [37]

Similarly, for reference in line 321 and 326. Use same reference style throughout the manuscript.

Response: We have corrected it according to the comment.  And carefully checked and corrected the references for the entire manuscript. (Line 345 348 352) 

Q12: Line 334: the abbreviation used for ‘Calcium oxide washing followed by combination of FA + SAEW (CaO-FA + SAEW)’ is different in the table and one mentioned below the table. Please correct it. This is applicable for all the tables.

Response: We have corrected it for all the tables. 

Q13: Table 3: Define the abbreviation ‘NaOCl’ in table text.

Response: NaOCl: Sodium hypochlorite. We added in the table text. We have added in the text of table1, 2, and 3. (Line 359, line 364, line 368)

Q14: Line 371: reference title is repeated twice.

Response: We have deleted the repeated word “reference”.

Q14: Figures: What was the detection limit for microbiological counts? Please mention them somewhere in the manuscript.

Response: Thanks for your valuable comments, the detection limit for microbiological counts is 1 log CFU/ fruit, we added the in the manuscript (line139) 

Reviewer 2 Report

In this version, the authors revised the manuscript, considering only a few of my questions and doubts. As in the previous version, the authors repeated some avoidable errors (citations, use of incorrect expressions such as “microflora”, references to not used methods (Gompertz), among others).

In the following section I am presenting some more issues that were not improved as suggested.

Introduction

Line 49- The citation is not correctly introduced in the text. Please avoid this kind of errors. The authors should be attentive when writing manuscripts and correcting them.

In line 54- The authors should improve the Introduction with results obtained with some more fruits such as “Rocha pear” as suggested in the previous revision.

Line 74- Calcium in capitals?

Material and methods

The authors should substitute the word “microflora” by “microbiota” in the entire manuscript, as suggested before. Microflora refers to plant origin microorganisms and bacteria do not belong to the Kingdom Plant.

Line 97- The authors should clearly describe the equipment, how it is constructed and their main parts.  Saying “self-developed” is not enough nor adequate for a scientific article.

Line 125- pay attention to the punctuation. Avoidable error. Correct in the entire manuscript.

Line 128- The description of the methods used to count yeast and mold and E. coli and coliforms is still not correct.

Line 137- Although the authors say in their revision that they have erased the “Gompertz model” from the manuscript, the expression was written in line 137.

Results and Discussion

Regarding the suggested improvement of the discussion, the authors did not present an explanation why the microbial reductions they obtained in the fruits tested are higher than the ones obtained in minimally processed fruits, as suggested in the revision letter sent to the authors. The authors did not try to improve their discussion as suggested by me.

Fig.4 and 5- In my opinion, using the techniques described in material and methods the detection limit for yeast and fungi is not 1 CFU/g. What was the volume inoculated and what was the countable dilution? They should correspond the counts of microorganisms inferior to the detection limit to 1 (loc CFU/g) and not 0.

References

The authors should verify the references as some mistakes are found for example reference 15 is equal to reference 38.

Author Response

 Response to the Reviewers comments

We are very thankful to you for your time and valuable suggestions on the manuscript. We appreciate the important comments and we are glad to respond to the comments highlighted by you.

The response to the point of comments are given below:

In this version, the authors revised the manuscript, considering only a few of my questions and doubts. As in the previous version, the authors repeated some avoidable errors (citations, use of incorrect expressions such as “microflora”, references to not used methods (Gompertz), among others).

In the following section I am presenting some more that were not improved as suggested.

Response: Thank you for valuable comments. We have carefully checked all manuscript and tried to improve the issues.

Introduction

Q1: Line 49- The citation is not correctly introduced in the text. Please avoid this kind of errors. The authors should be attentive when writing manuscripts and correcting them.

Response: We have corrected it according to your kind comment and carefully checked and corrected the citation for the entire manuscript. (Line 50)   

Q2: In line 54- The authors should improve the Introduction with results obtained with some more fruits such as “Rocha pear” as suggested in the previous revision.

Response: We have improved the introduction by adding more fruits such as “mangoes, rocha pear, and apples” related to decontamination using AEW as you kind suggested (Lines 50 to 58)

“David Santo et al. tested the antibacterial activity of AEW in the inhibition of Escherichia coli and Cronobacter sakazakii on fresh-cut mangoes, the results showed that AEW resulted in declining of E. coli (1.96 log CFU/g) and C. sakazakii (1.76 log CFU/g) populations [15]. Since fresh-cut fruits were a suitable substrate for the survival and growth of foodborne pathogens, the effect of AEW on foodborne bacteria population of fresh-cut fruits were detected though several studies, such as E. coli, Salmonella enterica and Listeria spp inoculated on ‘Rocha’ fresh-cut pears decreases values of 0.53–1.1 log CFU/g were achieved by EW (100 mg/L of free chlorine) washings [16]. Besides, AEW at 50 and 100 mg/L of free chlorine were used to treatment apple slices inoculated with E. coli, Listeria innocua or Salmonella choleraesuis and significantly decreased the populations of pathogen, when compared to that of sodium hypochlorite solution and distilled water [17].”

Reference:

[15]. Santo, D.; Graça, A.; Nunes, C.; Quintas, C. Escherichia coli and Cronobacter sakazakii in ‘Tommy Atkins’ minimally processed mangos: Survival, growth and effect of UV-C and electrolyzed water. Food Microbiol. 2018, 70, 49-54.

[16]. Graça, A.; Santo, D.; Quintas, C.; Nunes, C. Growth of Escherichia coli, Salmonella enterica and Listeria spp., and their inactivation using ultraviolet energy and electrolyzed water, on ‘Rocha’ fresh-cut pears. Food Control. 2017, 77, 41-49.

[17]. Graca, A.; Abadias, M.; Salazar, M.; Nunes, C. The use of electrolyzed water as a disinfectant for minimally processed apples. Postharvest biol tec. 2011, 61, 172-177.

Q3: Line 74- Calcium in capitals?

Response: We corrected it and carefully checked and corrected this kind of errors in the entire manuscript.  (Line 78)

Material and methods

Q4: The authors should substitute the word “microflora” by “microbiota” in the entire manuscript, as suggested before. Microflora refers to plant origin microorganisms and bacteria do not belong to the Kingdom Plant.

Response: We substituted the word “microflora” by “microbiota” in the entire manuscript as you suggested.

Q5: Line 97- The authors should clearly describe the equipment, how it is constructed and their main parts.  Saying “self-developed” is not enough nor adequate for a scientific article.

Response: We have tried to describe the construction of the electrolyzed water generator. (Line 99-100)

Q6: Line 125- pay attention to the punctuation. Avoidable error. Correct in the entire manuscript.

Response: Thank you so much for the comment .we carefully checked and corrected the punctuation in the entire manuscript. 

Q7: Line 128- The description of the methods used to count yeast and mold and E. coli and coliforms is still not correct.

Response: We corrected it in the text according the comments.

Dichloran Rose Bengal Chloramphenicol (DRBC Difco) agar were used to enumerate the yeast and mold and petri film (Difco) were used to enumerate the total coliforms, respectively. (Line 134, 138)

Q8: Line 137- Although the authors say in their revision that they have erased the “Gompertz model” from the manuscript, the expression was written in line 137.

Response: We corrected this sentence

 “The growth date of TAB, yeast and mold, and total coliforms during storage were monitored.”(Line 148)

Results and Discussion

Q9: Regarding the suggested improvement of the discussion, the authors did not present an explanation why the microbial reductions they obtained in the fruits tested are higher than the ones obtained in minimally processed fruits, as suggested in the revision letter sent to the authors. The authors did not try to improve their discussion as suggested by me.

Response: We would like to thank you for your time and suggestions. We have modified the Results and Discussion according to your kind comments.

We have given the explanation why the microbial reductions they obtained in the fruits tested are higher than the ones obtained in minimally processed fruits in the manuscript.

 “This is in agreement with the results presented by other researchers that SAEW, FA, and CaO were high effective on the bacterial reduction as chemical sanitizers [18, 24, 29].” (Line 220-221)

“The results presented indicate that FA, SAEW can inhibit the growth of TAB on the surface of apple, mandarin, and tomato during storage at 4 °C.” (Line268)

“The higher reduction of coliform populations on the surface of mandarin may be explained by the fact that food surface properties such as hydrophobicity, electric charge and roughness may influence the adhesion of microbial [16].(Line 242-244)

We have improved the discussion of the results comparing the microbial reductions they obtained with the microbial reductions described in other articles as your kind comments.

“The difference between two results  may be explained by  the instability of SAEW when exposed to air and light which may influence its effective on the bacterial reduction [14].”(Line 233-234)

“The results is in agreement with Daniel Rico et al. which the panelist considered acceptable all the lettuce samples treated with EW (12, 60, 120 mg/L of free chlorine) during 7days storage [37].”(Line 324-326)

“Several studies reported that EW did not affect the quality parameters of fresh production. In a study conducted by Thi-Van Nguyen et al. EW (20, 60 mg/L of free chlorine) was effective as a disinfectant for fresh-cut baby spinach, and remained above acceptable levels  over 13 days of storage at 4 ℃ [39].”(Line 326-329)

“The results described in this study are different to previous research regarding reductions of E. coli on fruit. Danyluk et al. [40] reported that 200 ppm free chlorine wetting stem scar of grapefruit resulted in 4.93 log CFU/ grapefruit reductions of E.coli on the surface of grapefruit. The higher reduction of E.coli may be explained by that higher available free chlorine concentration of SAEW used in their study. Santo et al. [15]” (line 344-348)

Reference:

[14]. Ding, T.; Ge, Z.; Shi, J.; Xu, Y.T.; Jones, C.L.; Liu, D.H. Impact of slightly acidic electrolyzed water (SAEW) and ultrasound on microbial loads and quality of fresh fruits. LWT-Food Sci. Technol. 2015, 60, 1195-1199.

[15]. Santo, D.; Graça, A.; Nunes, C.; Quintas, C. Escherichia coli and Cronobacter sakazakii in ‘Tommy Atkins’ minimally processed mangos: Survival, growth and effect of UV-C and electrolyzed water. Food Microbiol. 2018, 70, 49-54.

[16]. Graça, A.; Santo, D.; Quintas, C.; Nunes, C. Growth of Escherichia coli, Salmonella enterica and Listeria spp., and their inactivation using ultraviolet energy and electrolyzed water, on ‘Rocha’ fresh-cut pears. Food Control. 2017, 77, 41-49.

[18]. Tango, C.N.; Khan, I.; Kounkeu, P.F.N.; Momna, R.; Hussain, M.S.; Oh, D.H. Slightly acidic electrolyzed water combined with chemical and physical treatments to decontaminate bacteria on fresh fruits. Food Microbiol. 2017, 67, 97-105.

[24]. Chen, J.; Xu, B.; Deng, S.G.; Huang, Y.T. Effect of combined pretreatment with slightly acidic electrolyzed water and botanic biopreservative on quality and shelf life of bombay duck (Harpadon nehereus). J. Food Qual. 2016, 39, 116-125.

[29]. Sawai, J. Antimicrobial Characteristics of Heated Scallop Shell Powder and Its Application. Biocontrol Sci. 2011, 16, 95-102.

[31]. Wang, J.; Chen, J.; Hu, Y.; Hu, H.; Liu, G.; Yan, R. Application of a predictive growth model of pseudomonas spp. For estimating shelf life of fresh agaricus bisporus. J Food Prot. 2017, 80, 1676-1681.

[38].        Nguyen, T.-V.; Ross, T.; Van Chuyen, H. Evaluating the efficacy of three sanitizing agents for extending the shelf life of fresh-cut baby spinach: food safety and quality aspects. Renew agr food syst. 2019. 4(2): 320–339.

[40].            Danyluk, M.D.; Friedrich, L.M.; Dunn, L.L.; Zhang, J.X.; Ritenour, M.A. Reduction of Escherichia coli, as a surrogate for Salmonella spp., on the surface of grapefruit during various packingline processes. Food Microbiol. 2019, 78, 188-193.

Q10: Fig.4 and 5- In my opinion, using the techniques described in material and methods the detection limit for yeast and fungi is not 1 CFU/g. What was the volume inoculated and what was the countable dilution? They should correspond the counts of microorganisms inferior to the detection limit to 1 (loc CFU/g) and not 0.

Response:

We have revised this part in the text according to the comments.

“1mL of sample suspension containing natural microbiota were mixed with Tryptone Soy agar (TSA Difco) and poured into plate to enumerate the total aerobic bacteria (TAB). Dichloran Rose Bengal Chloramphenicol (DRBC Difco) agar were used to enumerate the yeast and mold. Then 1mL of sample suspension were 10-fold diluted with 0.1% buffered peptone water (BPW Difco). Population of cell in each culture was confirmed by plating 0.1 mL serial dilution on the agar plates and incubating at 37°C for 24 h. Population of total coliforms was confirmed by plating 1 mL serial dilution on the petri film. Colonies were enumerated and expressed as log CFU/fruit, and the detection limit for microbiological counts is 1 log CFU/ fruit.”(Line 133-140)

 Q11: Fig.3. 4 and 5 we correspond the counts of microorganisms inferior to the detection limit to 1 (loc CFU/fruit).

Response: We have corrected the Fig.3. 4 and 5, Correspond the counts of microorganisms inferior to the detection limit to 1 (log CFU/fruit).

Q12: References

The authors should verify the references as some mistakes are found for example reference 15 is equal to reference 38.

Response: Thank you so much for valuable comments. We deleted the repeated reference 38 in the citation part, and corrected the number in the text. 

Reviewer 3 Report

Page 16 Line 371:  References  (delete it because it was repeated)

Author Response

Q1: Page 16 Line 371:  References (delete it because it was repeated)

Response: Thank you so much for your time and valuable comments. We deleted the repeated reference as suggested by you and carefully checked and corrected this kind of errors in the entire manuscript.